# A Case of Adaptive Laboratory Evolution (ALE): Biodegradation of Furfural by *Pseudomonas pseudoalcaligenes* CECT 5344

**DOI:** 10.3390/genes10070499

**Published:** 2019-06-29

**Authors:** M. Isabel Igeño, Daniel Macias, Rafael Blasco

**Affiliations:** 1Departamento de Bioquímica y Biología Molecular y Genética, Facultad de Veterinaria, Universidad de Extremadura, 10003 Caceres, Spain; 2Meat and Meat Products Research Institute (IProCar), BioMic Research Group, Universidad de Extremadura, 10003 Caceres, Spain

**Keywords:** *Pseudomonas*, bioethanol, furfural, ALE, AraC

## Abstract

*Pseudomonas pseudoalcaligenes* CECT 5344 is a bacterium able to assimilate cyanide as a nitrogen source at alkaline pH. Genome sequencing of this strain allowed the detection of genes related to the utilization of furfurals as a carbon and energy source. Furfural and 5-(hydroxymethyl) furfural (HMF) are byproducts of sugars production during the hydrolysis of lignocellulosic biomass. Since they inhibit the yeast fermentation to obtain bioethanol from sugars, the biodegradation of these compounds has attracted certain scientific interest. *P. pseudoalcaligenes* was able to use furfuryl alcohol, furfural and furoic acid as carbon sources, but after a lag period of several days. Once adapted, the evolved strain (R1D) did not show any more prolonged lag phases. The transcriptomic analysis (RNA-seq) of R1D revealed a non-conservative punctual mutation (L261R) in BN5_2307, a member of the AraC family of activators, modifying the charge of the HTH region of the protein. The inactivation of the mutated gene in the evolved strain by double recombination reverted to the original phenotype. Although the bacterium did not assimilate HMF, it transformed it into value-added building blocks for the chemical industry. These results could be used to improve the production of cost-effective second-generation biofuels from agricultural wastes.

## 1. Introduction

The development of renewable resources is one of the key actions to palliate climate change, which is largely a consequence of the world’s dependence on petrol. On the other hand, contamination of the environment is an inevitable consequence of human development. These are global problems that need international agreements [1]. Biotechnology can offer solutions to these challenges, such as the production of bioethanol as a substitute to gasoline [2]. Biotechnology can also offer alternatives to the physical-chemical treatment of contaminating compounds, either by avoiding their production, or by mitigating their impact once it has occurred. The biodegradation of pollutants is, in general, a technology that has good social acceptance [3]. *Pseudomonas pseudoalcaligenes* CECT 5344 was isolated from sludge of Guadalquivir River, and it is able to use cyanide as the only source of nitrogen [4]. Cyanide is an extremely toxic compound used in the synthesis of organic compounds such as nitriles, plastics, paints, adhesives, cosmetics, etc., while mining activities and the jewellery industry are the main source of cyanurated wastes [5,6,7,8]. This strain tolerates an unusually high concentration of cyanide (up to 30 mM) [4], but it requires a suitable carbon source for growing. The sequencing of the genome of *P. pseudoalcaligenes* CECT 5344 has made it possible to predict which carbon sources can be used by this bacterium, such as the assimilation of furanic compounds [9]. Furfurals are aromatic natural compounds formed by the dehydration of sugars during the thermochemical pre-treatment of the lignocellulosic materials to release fermentable sugars. The production of biofuels from lignocellulosic residues, which is part of the so-called second-generation biofuels, constitutes a viable option for reducing the greenhouse effect and for providing an alternative to fossil fuels [10,11]. There are different pre-treatment technologies of lignocellulosic residues. One of the parameters that has to be taken into account to optimize the process is avoiding the formation of potentially inhibitory compounds to the posterior yeast fermentation process [12]. From the food technology perspective, furfurals are potential carcinogenic compounds used as a marker of honey adulteration, generated by acid-catalyzed dehydration of carbohydrates of food-containing sugars [13]. In any case, furfural (F), or fufuralaldehyde, and 5-hydroxymethyl furfural (HMF) are natural products that can be eliminated by using the capacity of some microorganisms to metabolize them [14,15,16,17,18,19,20,21,22]. Other furanic derivatives are furoic acid (FA) and furfuryl alcohol (FFA), all of them with the common thread of having an aromatic furan ring. The variety of furanic compounds degrading species is limited mostly to Gram-negative aerobic bacteria and some Gram positives [17], with a few exceptions including fungi [14]. In the first degradation route currently proposed, furfural is oxidized to 2-furoic acid (FA), which is subsequently transformed into 2-oxoglutarate, a Krebs cycle intermediate [23]. The complete metabolic pathway for the assimilation of F and HMF, as well as the genetic of the process, was first described in the soil isolate *Cupriavidus basilensis* HMF14 [24] (Figure 1). In this strain, the *hmfABCDE* gene cluster is responsible for the assimilation of furoic acid. The first reaction in the pathway is catalysed by the 2-furoyl-CoA synthetase (HmfD), producing 2-furoyl-CoA from 2-furoic acid. The conversion of 2-furoyl-CoA is into 5-hydroxy-2-furoyl-CoA in *C. basilensis*; HMF14 is catalysed by the molybdenum-dependent 2-furoyl-CoA dehydrogenase (HmfABC). The final steps of the proposed furoic-acid metabolic pathway consist of the transformation of 5-oxo-2-furoyl-CoA into 2-oxoglutarate. No gene has been assigned to the hydrolysis of the lactone, whereas *hmfE* has been proposed to encode a specific 2-oxoglutaroyl-CoA thioesterase [24] (Figure 1). *P. pseudoalcaligenes* contains an *hmfABCDE* gene cluster homologous to the gene cluster shown to be essential for the assimilation of furfural in *C. basilensis* HMF14 (Figure 1). Concretely, the amino acid sequence of HmfA from *C. basilensis* HMF14 (GenBank ADE20399.1) is 64% identical to the homologous protein of *P. pseudoalcaligenes* (BN5_2298, 76% positives). The % identity/% similarity for the rest of the proteins are: 59%/72%, 77%/83%, 61%/75% and 80%/88%, for HmfB (GenBank ADE20400.1), HmfC (GenBank ADE20401.1), HmfD (GenBank ADE20402.1) and HmfE (GenBank ADE20403.1), respectively. Moreover, this locus also contains downstream *hmfABCDE,* a gene (*benE*) belonging to the Major Facilitator Superfamily (MSF)-family of transporters and two separate genes homologous to genes related to the assimilation of furfural in *Pseudomonas putida* Fu1 [9,25] (Figure 1). One of them belongs to the AraC-family of regulators. AraC from *P. putida* Fu1 (GenBank ACA09742.1) is 75% identical (88% similar) to its orthologous gene product in *P. pseudoalcaligenes* (BN5_2307, Figure 1). The other upstream gene (*PsfD*) codes for a putative conserved protein usually annotated as maturation factor for molybdenum containing dehydrogenases, like xanthine and CO dehydrogenases [9,25]. In that respect, the furoyl-CoA dehydrogenase was proposed to be a molydo-protein [26]. PsfD protein from *P. putida* Fu1 (GenBank ACA09741.1) is 81% identical (89% similar) to PsfD form *P. pseudoalcaligenes* (BN5_2306, Figure 1). To date, the architecture of this operon presented in Figure 1B has not been described in this context. Here we show that this operon is functional after an adaptation period ending up with the selection of a punctual mutant in the *araC*-type regulator. Therefore, the locus described here seems to be a hybrid furfural assimilating system containing horizontally transferred genes homologous to the catalytic genes for the assimilation of FA described in *C. basilensis* [24] and the regulatory and accessory genes described in *P. putida* Fu1 [25]. 

## 2. Materials and Methods

### 2.1. Bacterial Strains, Media and Growth Conditions

Strain R1 is a spontaneous mutant of *P. pseudoalcaligenes* CECT5344 [27] resistant to rifampicin (40 μg/mL). *P. pseudoalcaligenes* CECT5344 strain R1D was obtained after four serial transfers of *P. pseudoalcaligenes* CECT5344 R1 to a M9 medium with furfural (10 mM) as the sole carbon source. A dilution 1:100 of the previously grown preculture was used as inoculum. For the rest of the growth curves, unless otherwise stated, the inoculum was the equivalent of an overnight culture diluted 1:10 into fresh medium. Bacterial growth was monitored by measuring the absorbance at 600 nm. Cells were grown in either minimal medium (M9) adjusted to pH 8.5 [4] or in LB medium [28] adjusted to pH 8.5. Cell cultures were prepared in Erlenmeyer flasks filled with 1/10 (v/v) of their nominal volume in order to ensure aerobic conditions and incubated on a rotatory shaker at 190 rpm and 30 °C. For minimal medium, ammonium chloride (5 mM) was used as the nitrogen source and 4 g/L of sodium acetate, furfural (10 mM), furfuryl alcohol (10 mM) or furoic acid (10 mM) were added as the sole carbon source. *Escherichia coli* XL1 *blue* MRF′ cells (Stratagene, Agilent Technologies, Santa Clara, CA, USA) were grown aerobically at 37 °C in complex LB medium [28] with ampicillin (100 μg/mL). Where appropriate, the following compounds were added to the media: X-Gal (5-bromo-4-chloro-3-indolyl-β-d-galactopyranoside, 0.2 mM, Appli-Chem (Barcelona, Spain), IPTG (isopropyl-β-d-1-thiogalactopyranoside, 0.5 mM, Sigma-Aldrich (St. Louis, MO, USA). Electrocompetent cells were prepared by the growth of cultures up to an optical density at 600 nm (OD_600_) of 0.35 and centrifugation for 20 min at 1000× *g*, followed by three successive washes (4 °C) in 1:1, 1:2, 1:50, and 1:500 volumes of 10% glycerol. The last solution, which yields the stock of electrocompetent cells, contained yeast extract (0.125%) and tryptone (0.25%). A mixture of 50 μL of cells (2 × 10^10^ to 3 × 10^10^ CFU/mL) and 1 to 10 ng of DNA was electroporated in 2-mm cuvettes with a Bio-Rad Gene Pulser II apparatus, (Bio-rad, Hercules, CA, USA) operated at 2.5 kV, 25 μF, and 200 Ω (4- to 5-ms time constants).

### 2.2. Preparation of Cell-free Extracts

The cells of *P. pseudoalcaligenes* CECT 5344 R1D grown with furfuryl alcohol as a carbon source (500 mL culture) were collected by centrifugation at the end of the logarithmic phase and resuspended in 50 mM Tris-HCl (pH 8), containing a complete protease inhibitor cocktail, as recommended by the supplier (Roche, Penzberg, Germany) and glycerol (10%). The cells were disrupted by two passages through a French pressure cell operated at 130 MPa and the cell debris removed by centrifugation at 18,000× *g* for 15 min.

### 2.3. Enzymatic Assays

Furfural dehydrogenase (FDH) and furfuryl alcohol dehydrogenase (FFADH) were assayed as previously described for *C. basilensis* [24], but optimizing the assay for *P. pseudoalcaligenes* (pH and temperature). FFADH (E.C.1.1.1.--) was assayed spectophotometrically at 65 °C and pH 9.5 (50 mM Tris/phosphate/carbonate) following the increment of absorbance at 340 nm due to the production of NADH (ε = 6220 M^−1^·cm^−1^). The reaction mixture contained NAD^+^ (1.5 mM), FFA (5 mM) and the appropriate amount of cell-free extract (50–100 μL cell-free extract, 0.5–1 mg protein, approximately) in a final volume of 1 mL. FDH (E.C. 1.2.3.1) was measured spectrophotometrically at 65 °C and pH 6.5 (50 mM Tris/phosphate/carbonate) following the increment of absorbance at 522 nm due to the reduction of the artificial electron acceptor DCPIP (ε = 21,000 M^−1^·cm^−1^). The reaction mixture contained, in a final volume of 1 mL, 0.33 mM PMS, 0.1 mM DCPIP, 5 mM F, and the appropriate amount of enzyme (50–100 μL cell-free extract, 0.5–1 mg protein, approximately).

### 2.4. Chromatographic Separation of FDH and FFADH

Cell-free extract from FFA-grown cells was loaded into an anion exchange chromatography (mono Q 5/50 GL, GE Healthcare (Chicago, IL, USA) attached to an Akta Purifier, GE. All the chormatographies were carried out at 4 °C. The column was equilibrated in buffer A (Tris/HCl 50 mM pH 8, 2 mM DTT and glycerol (2%)). The cell-free extract (2 mL) was loaded into the column at a flow rate of 1 mL/min. The unbound protein was washed with 12.5 mL of buffer A. Then, it was applied at a gradient of 1 mL from 0 to 0.1 M NaCl, always in the same buffer, which was maintained as isocratic during 5 mL, followed by a 20 mL gradient from 0.1 to 0.5 M NaCl and 0.5 to 1 M during 10 mL. Finally, the column was regenerated with 5 mL of buffer A containing 2 M NaCl. The fractions (0.5 mL) were analyzed for the presence FFADH and FDH activities.

### 2.5. Analytical Methods

#### 2.5.1. HPLC

The concentration of furanic intermediates was determined as follows. The supernatants of the culture media were obtained by centrifugation of 1.0 mL of bacterial culture in a microfuge at maximal speed (14,000 rpm) for 10 minutes at room temperature. The supernatants were further filter-sterilized using a 0.2-µm filtration unit and stored at −20 °C. Concentrations of furan derivatives were determined from sample supernatants by high-performance liquid chromatography (HPLC) on a HPLC System Gold (Beckman) system. The column used was an Anion Exchange ION-300 (ICE-99-9850) (300 × 7.8 mm, Transgenomic, Omaha, NE, USA) operated at 65 °C. As eluent H_2_SO_4_ 5 mN was used at a flow of 0.6 mL·min^−1^. Furfural and furoic acid were detected at 278 nm and furfuryl alcohol at 217 nm. 

*Glucose concentration* was colorimetrically measured by using a commercially available enzymatic test based on the glucose/peroxidase activities (Biosystems, Barcelona, Spain).

*Protein concentration* was determined by the Bradford procedure [29].

#### 2.5.2. DNA Manipulation

DNA was sequenced using services provided by Sistemas Genómicos (Valencia, Spain). Genomic DNA was purified from *P. pseudoalcaligenes* CECT5344 cells grown in liquid LB medium at pH 8.5 using the GNome DNA isolation kit (QBIOgene). Plasmid DNA was purified with the Genopure plasmid Midi Kit (Roche) from *E. coli* XL1 *blue* MRF′ cells grown in liquid cultures of LB media supplemented with the antibiotic used for selection. The *E. coli* XL1 *blue* MRF′ strain was used to clone recombinant DNA, performing restriction enzyme digestion and ligation as recommended by the manufacturers (Fermentas and Promega, respectively). Plasmids were introduced into *E. coli* XL1 *blue* MRF′ and *P. pseudoalcaligenes* CECT5344 cells by electroporation, as described previously [27]. Mutagenesis of *edd* gene (BN5_3048): To amplify by PCR a section of the *edd* ORF based on the DNA sequence of the *P. pseudoalcaligenes* CECT5344 (GenBank accession no. JN408065), two couples of sets of specific primers flanking the *edd* gene were designed (edd_9_U /edd_730_L and edd_1140_U /edd_1737_L, Table 1). A BamHI restriction site was incorporated into edd_730_L and edd_1140_U primers. The amplified fragments were cloned separately into pGEM-T Easy vector (Promega) generating the p*edd*1 and the p*edd*2 plasmids. The p*edd*2 plasmid was linearized with ApaI and BamHI. The plasmid p*edd*1 was digested with the same enzymes and the resulting fragment was gel-purified and subcloned into p*edd*2 to generate p*edd*1-2, thus obtaining a PGEM-TE derivative plasmid with two internal sequences of *edd* gene separated by a BamHI restriction site. On the other hand, a 1.0 kb BamHI fragment from the pMS255 plasmid [30] containing the gentamicin resistance gene (*aacC1*) was cloned into BamHI-digested PGEM-TE plasmid containing the two internal *edd* fragments (Appendix A). Finally, the resulting suicide plasmid was transferred to *P. pseudoalcaligenes* CECT5344 R1D by electroporation. The mutants were selected on gentamicin (10 µg/mL, Sigma-Aldrich) and mutant strains resulting from double homologous recombination were isolated. The insertion of the gentamicin resistance gene and the loss of the plasmid backbone were confirmed by PCR. Mutagenesis *araC* gene (BN5_2307): Based on the DNA sequence of the *P. pseudoalcaligenes* CECT5344 (GenBank accession no. JN408065), one couple of specific primers (araC_157_U/araC*_823_*L, Table 1) flanking the *ara*C gene was designed for amplification of a section of the *ara*C ORF by PCR. PCR was performed and the amplified fragment that contained a KpnI restriction site at position 542 was cloned into pGEM-T Easy vector (Promega), thus obtaining a PGEM-TE derivative plasmid with one internal sequence of *ara*C. The plasmid was digested with KpnI and ligated to a 1.0-kb KpnI fragment containing the gentamicin-resistance gene (*aac*C1) from the pMS255 plasmid [30]. (Appendix A). The resulting plasmid was transferred to *P. pseudoalcaligenes* CECT5344 R1D by electroporation. The mutants were selected on gentamicin (10 µg/mL, Sigma-Aldrich), and mutant strains resulting from double-homologous recombination were isolated. The insertion of the gentamicin resistance gene and the loss of the plasmid backbone were confirmed by PCR.

#### 2.5.3. PCR Reaction Conditions

PCR samples were prepared with the following components: 0.5 ng/µL DNA, 1.0 µM of each primer, 2.0 mM MgCl_2_, 0.2 mM each deoxynucleoside triphosphate (dNTP), 0.5 U Taq DNA polymerase, and 2 µL buffer as recommended by the manufacturer (Biotools), in a final volume of 20 μL. The PCR conditions were 2 min at 95 °C; followed by 30 cycles of 20 s at 95 °C, 10 s at 65.6 °C (*edd*) or 64 °C (*ara*C) and 1 min at 72 °C, followed by a 5 min extension at 72 °C. 

## 3. Results

*P. pseudoalcaligenes* CECT5344 R1 was able to utilize furfural (up to 40 mM), furoic acid (up to 20 mM), and furfuryl alcohol (up to 20 mM) as the sole carbon and energy source, although after a very long lag phase (Figure 2A). The longest lag phase was observed with furfural as a C-source (Figure 2(A2)). In fact, furfural concentrations higher than 5 mM increased the lag phase, thus suggesting that this compound is toxic at a high concentration. Nevertheless the tolerance of *P. pseudoalcaligenes* CECT 5344 to furfural is relatively high (up to 40 mM) if compared to *C. basilensis* HMF14 (up to 12 mM) [16]. This toxic effect was not observed for furoic acid (FA), and furfuryl alcohol (FFA) was not toxic up to a concentration of 20 mM (Figure 2A). The maximum cell growth increased with the concentration of the furanic compound up to a concentration of 20 mM, indicating that below this concentration the growth was carbon limited. It is remarkable that at the same concentration of furanic compound, the maximum growth was highest with FFA, followed by F and finally FA. This is in agreement with the chemical compositions of these compounds, the alcohol being the most reduced, followed by the aldehyde and then the carboxylic acid. Both F and FFA were, in a first instance, almost stoichiometrically transformed into furoic acid (Figure 2B), thus indicating that the pathway for the assimilation of this compound is not active in the wild type strain of *Pseudomonas pseudoalcaligenes* CECT 5344 R1. Only after several days of incubation, a clear increase in cell growth was observed, which was concomitant with the assimilation of FA (Figure 2B). 

*P. pseudoalcaligenes* CECT 5344 R1 failed to grow on 5-(hydroxymethyl)furfural (HMF), even after a prolonged incubation period of more than 10 days (Figure 3A). The analysis of the culture media by HPLC revealed that HMF was completely exhausted after 25 h, indicating that although this bacterial strain was unable to assimilate HMF, it had the capability of transforming it. Two new compounds were detected in the couture media after consumption of HMF, 2,5-furandicarboxylic acid (FDCA) in a proportion below 5% of the HMF consumed and an unknown aromatic compound (Figure 3A). In *C. basilensis* HMF14, HmfH catalyzes the oxidation of 5-hydroxymethyl-2-furoic acid (HMFA) to FDCA, whereas HmfFG catalyzes the decarboxylation of FDCA to FA [24]. Therefore, in this strain, the metabolic pathways for the assimilation of HMF, FFA and F converge in FA. In the genome of *P. pseudoalcaligenes* CECT 5344 R1, no homologous genes to *hmfH* and *hmfFG* of *C. basilensis* were observed. This genotype agrees with the fact that this bacterium does not assimilate HMF. Since HMFA is the substrate of HMFH, the unknown compound accumulated in the culture media from HMF (Figure 3A) could be HMFA. To test if the incapacity of *P. pseudoalcaligenes* was due to a problem of induction, the bacterium was inoculated in media containing both F and HMF (Figure 3B). The result was that the bacterium exclusively used F and that the presence of HMF resulted in being toxic (Figure 3B). Therefore, the absence of catalytic enzymes for the assimilation of FA was not the reason for the inability of *P. psedoalcaligens* to assimilate HMF, but the absence of reactions connecting HMFA and FDCA to FA. The transformation of FFA into FA takes place in two consecutive oxidative steps. In *Pseudomonas putida* Fu1, two different and inducible enzymes catalyze these reactions [23], but in *C. basilensis*, although no concrete genes have been assigned, it could be possible that the same enzyme catalyzes both oxidations [24]. In *P. pseudoalcaligenes* CECT5344 R1, both dehydrogenase activities (FFADH and FDH in the scheme of Figure 1D) co-eluted after anion exchange chromatography (not shown), thus suggesting that it is the same enzyme. Again, both activities had the same optimum temperature at 65 °C. Furfural dehydrogenase activity was clearly induced by furfuryl alcohol, if compared with acetate, FA or LB medium (not shown). Although these results suggest that the same enzyme could catalyze the oxidation of FFA to F and of F to FA, the only clear conclusion is that transformation of FFA to FA and conversion of FA to 2-oxoglutaric acid takes place through different pathways.

Once grown on furfural as a C-source, the serial dilutions of a culture of *P. pseudoalcaligenes* CECT 5344 R1 spread on solid minimal medium with furfural or FA giving colonies with two morphologies, big or small colonies. The small colonies had the same phenotype as the original stain (R1), but the big colonies grew faster on furfural even after successive generations on non-selective medium (LB). In fact, after four serial re-inoculations of the bacteria in in fresh media with furfural (10 mM) as the sole carbon source, allowed the selection of a mutant (big colonies in FA plates) in which the growth lag phase was drastically reduced and had a reproducible growth rate of 0.29 h^−1^ on furfural (5 mM). Figure 4A illustrates the isolation of the mutant, thereafter called R1D, and its phenotype in comparison to the *wt*. The selection of mutants with improved capacities has been widely used in biotechnological processes for the selection of evolved phenotypes, although having poor knowledge of their underlying genotype [31], also in the context of furfural assimilation [32]. For example, *P. putida* S12 expressing the *hmfABCDE* genes from *C. basilensis* HMF14 strain, was only able to efficiently assimilate furfural when adapted by repeated inoculation in media with furfural. The same has been recently described for *Pseudomonas putida* KT2440 expressing a 12 kb DNA fragment containing the *hmf* gene cluster from *Burkholderia phytofirmans* [33]. The next generation sequence (ngs) techniques open the possibility to analyze the genotypic variation associated with the evolved phenotypes. Adaptive laboratory evolution (ALE) takes profit of this advantage and may have a tremendous application in the rational design of genetically manipulated microorganisms, as well as in understanding some basic evolving mechanisms of living beings [31,34,35]. In our laboratory, the transcriptomic analysis of the R1D mutant in response to furfuryl alcohol was analyzed and it revealed to be more complex than expected (not shown). Interestingly, the transcriptomic reads sequences (RNA-seq) were obtained from the mutant strain (R1D), whereas the genome sequences available are from the wt strain [36]. The comparison of both sequences in the *hmf* locus revealed that the R1D mutant had a point mutation in a possible regulatory gene of the *araC* family. The presence of the mutation was confirmed by re-sequencing the *araC* gene (BN5_2307) (Figure 4B). The observed point mutation was a transversion (782T>G), leading to the non-conservative change of the triplet CTT (Leu) to CGT (Arg) (Figure 4B). The alignment of several AraC proteins revealed that the Arg in position 261 is a conserved residue in most members of this family of regulators (not shown). This position is located in the HTH domain of the protein and it is also conserved in *E. coli* [37], thus suggesting its essentiality. In conclusion, it seems that the pathway for the assimilation of FA is not active in the wt strain of *P. pseudoalcaligens* because AraC is not functional, and that the L262R mutation generates the active and functional regulator (AraC*) in the evolved R1D strain. In order to check this hypothesis, a mutant of the *araC** gene of *P. pseudoalcaligenes* CECCT 5344 R1D was generated by double recombination. As expected, the mutant strain was unable to use furfural as a C-source (Figure 4C).

In addition to furfural, the mutant R1D was also capable to assimilate furfuryl alcohol and furoic acid (Figure 5). However, this strain remained unable to use HMF as a carbon source.

It is evident that R1D grew faster than R1 and with similar rates to other furfural-degrading strains reported in the literature. Table 2 summarizes the growth parameters of R1 and R1D strains of *P. pseudoalcaligenes* CECT 5344 in comparison to reference strains. In addition to the shorter lag phases and higher growth rates, a notable difference between the wild strain (R1) and the R1D mutant was the accumulation of FA during the lag phase. As shown in Figure 2, the wt strain stoichiometrically transformed FFA and F to FA. By contrast, in the RD1 mutant, FA was the only transiently accumulated form F (Figure 5B), whereas it was hardly detectable in media with FFA (Figure 5A). No lag phase was observed in the R1D strain growing with FA. Therefore, the presence of FA immediately induces its assimilation in R1D, thus suggesting that FA could be the inducer of the process. It is evident that the difference between R1 and R1D relies on the process of assimilation of furoic acid, not in the oxidation of alcohol and aldehyde to furoic acid. In analogy with the assimilation of toluene and xylenes in *P. putida* mt-2 [38], we can divide the pathway in two segments, the upper pathway and the lower pathway. We can consider the upper pathway as the oxidation of the alcohol group (-CH_2_OH) and aldehyde (-CHO) to acid (-COOH). In the case of *C. basilensis*, the decarboxylase has to be included in the upper pathway, making the metabolism of HMF converge to FA. The lower pathway consists in the mineralization of furoic acid. 

It is worth noting that these experiments confirmed the fact that the maximal cell growth of *P. pseudoalcaligenes* with FFA was higher than with F followed by FA (Table 2). 

It is evident that R1D is an evolved strain much more efficient than the wt in the assimilation of FFA, F and FA. Nevertheless, the utilization of these capabilities for the elimination of the inhibitory compounds from lignocellulosic hydrolysates requires that the bacterium leave the sugars intact for their further fermentation process. *P. pseudoalcaligenes* CECT 5334 was unable to use as a C-source neither xylose, sucrose, arabinose, mannose, nor galactose. The capability of *P. pseudoalcaligenes* to use sugars was restricted to glucose. The chemical composition of the lignocellulosic hydrolysates depends on the raw material utilized and the treatment employed [39]. Therefore, the capacity of *P. pseudoalcaligenes* CECT 5344 R1D to use glucose and furfural simultaneously was studied. As expected, R1D assimilated both compounds simultaneously in blended media (Figure 6). From Figure 6, it became evident that F is toxic, since maintaining the concentration of glucose, the lag phase increased when furfural concentration increased (Figure 6A,B). In any case, since both glucose and F were assimilated simultaneously by *P. Pseudoalcaligenes*, the construction of a mutant impaired in glucose assimilation was designed. Most *Pseudomonas*, which have a relatively limited ability to assimilate sugars, usually assimilate glucose through the Entner-Doudoroff pathway [40], and the inactivation of the *edd* gene usually causes the inability to assimilate glucose [41]. As shown in Figure 6C, the *edd^−^* strain was as efficient as R1D assimilating furfural but glucose remains unaltered in the culture media. The *edd^−^* mutant was also able to use furfuryl alcohol and furoic acid as a C-source. 

## 4. Discussion

The biodegradation of contaminants and the production of renewable energies are among the most important challenges of modern society. Bioethanol is a sustainable fuel that can be obtained by the yeast fermentation of sugars [42]. When using sugars from lignocellulosic by-products instead of sugars from food crops, the product is called second-generation bioethanol. However, the hydrolysis of these polymers generates, in addition to fermentable sugars, compounds that act as inhibitors of the yeast catalyzing further fermentation. The formation of furfural from pentoses was described many years ago [43]. Furfurals, and specially HMF and its derivatives, are feedstock for the synthesis of numerous valuable products. Therefore, their production in a sustainable manner is receiving an increasing interest [44]. On the other hand, from the bioethanol-production point of view, it is a challenge eliminating these inhibitors leaving sugars intact, with the purpose of increasing efficiency in second-generation ethanol production. Furfural and 5-hydroxymethyl furfural (HMF) are not the only inhibitory compounds present in the hydrolysates, but many others are not so well characterized, such as methanol or acetate [39]. The selection of yeast resistant to aldehydes may partially circumvent the problem. In fact, yeasts resistant up to 40 mM furfural and 80 mM HMF have been described recently [45,46]. Nevertheless, and taking into account the diversity of hydrolysates, the direct elimination of inhibitory compounds has to be taken into account as an alternative (bio)technology. 

In this manuscript, we describe the selection of an adaptive mutant of *Pseudomonas pseudoalcaligenes* CECT 5344 able to assimilate furfural. The mutation was found to be located in a regulatory gene of the AraC/XylS family of activators [47]. In *E. coli, araC* is necessary for the assimilation of L-arabinose, a five carbon sugar [37]. Although speculative, perhaps it is not casual that the homologous gene in *P. pseudoalcaligenes* CECT 5344 could recognize as an activator the five-carbon dehydrated and oxidized pentose FA. This hypothesis has to be tested experimentally in the future. AraC homologous are proteins approximately 300 aa long, whose C-terminal segment is the HTH domain interacting with DNA. The mutation in the adapted strain was detected just in this domain. The substitution of a hydrophobic amino acid by a basic one (L262R) introduces a positive charge in the protein that seems to be essential for interacting with the negatively-charged DNA molecule. The less-conserved N-terminal domain of the AraC family is presumed to contain binding sites for specific activator molecules that confer specificity to each member [47]. 

Adaptive mutations frequently target regulatory genes [34], although there are interesting examples in which the genetic adaptation lies in catabolic genes. For example, in *P. putida* KT2440, a single point mutation was detected causing the suppression of a frameshift mutation in the transporter (*galT*), thus allowing the evolved strain to grow in gallic acid [48]. The *hmf* locus in *P. pseudoalcaligenes* contains two different modules, a catabolic one homologous to *C. baisilensis* HMF14 (plus *benE*) and another one homologous to *P. putida* Fu1 containing the regulatory gene *araC* (Figure 1B). As far as we know, no gene homologous to *benE* has been described in the context of FA assimilation. *benE* is a member of the MSF-family of transporters, homologous to the benzoate transporters. Since both compounds, benzoate and furoate, are aromatic monocarboxylic acids, it can be speculated that BenE is a furoate transporter, although this hypothesis needs further experimental evidence. The *hmf* locus (Figure 1) is flanked by mobile genetic elements, thus suggesting that it has been horizontally transferred. Furthermore, this hypothesis is in agreement with the sequence composition of the locus. The average GC content of the genes flanking the *hmf* locus in *P. pseudoalcaligenes* (green genes in the Figure 4B) is around 62%, which is also the %GC content of the genome of *P. mendocina* ymp (Figure 4A, 62.8%) and *P. pseudoalcaligenes* CECT 5344 itself (62.34%). By contrast, the average composition (%GC) of the *hmf* operon (BN5_2298 to BN5_2305) is 66.5%, close to the composition of *Cupriavidus basislensis* (65.3%) and other betaproteobacteria. The other two genes present in the island are BN5_2306 (*psfD*) and BN5_2307 (*araC*), whose %GC are 67.69% and 57.16%, respectively. It is also remarkable that all genes present in the *hmf* locus (Figure 1B) are singletons in relation to most Pseudomonaceae, except *psfD* and *benE*. In the evolution of prokaryotic metabolic networks and their regulation, the number of transcriptional regulators grows faster than the metabolic genes [49]. The horizontally transferred *hmf* pathway homologous to that described in *C. basilensis* HMF14 does not include its dedicated transcriptional regulator, but it seems that it has been acquired in a separate module from a second donor strain. Curiously, the regulator was originally in an inactive form, but evolved to the active form under selective pressure (Figure 4).

Even though *P. pseudoalcalignes* CEC T 5344 R1D assimilates furfural very efficiently, it could not be useful for the pre-treatment of lignocellulosic residues because it simultaneously assimilates glucose. By contrast, the *edd*^−^ mutant assimilated furfural leaving the glucose intact (Figure 6). This is an important feature in comparison to the equivalent mutant in *P. putida* KT2440, that accumulates 6-phosphoogluconate from glucose [40]. On the other hand, *P. pseudoalcaligens* could not assimilate HMF (Figure 3), the other main inhibitory component of hydrolysates. In fact, the genome analyses anticipated this result due to the absence of homologous genes to *hmfH* and *hmfFG* genes. These genes in *C. basilensis* code for the enzymes catalyzing the oxidation of HMFA to FDCA and the decarboxylation of the latter to generate furoic acid, respectively. FA is the metabolite in which converges the assimilation of HMF and furfural in *C. basilensis* HMF14 [24]. To circumvent the problem, it would be convenient to have bacteria capable of eliminating HMF. We have isolated, by selective enrichment from ashes, some bacterial strains belonging to the genus *Pseudomonas*, which are able to assimilate HMF in addition to furfural, furoic acid or furfuryl alcohol [50]. These abilities could be used in mixed cultures, provided that they do not assimilate sugars, or by constructing the required traits using the available genetic modules. Even though *P. pseudomonas* CECT 5344 R1D fully transformed HMF into two new compounds, 5-Hydroxymethylfuranoic acid (HMFA) and 2,5-furandicarboxylic acid (FDCA), both of which are versatile chemical intermediates of high industrial potential [51,52]. Therefore, the *edd*^−^ mutant of the evolved R1D strain of *P. pseudoalcaligenes* described in this manuscript may increase the productivity of second generation bioethanol both by eliminating yeasts’ inhibitory chemicals and by producing value-added chemicals from biomass. 

## Figures and Tables

**Figure 1 genes-10-00499-f001:**
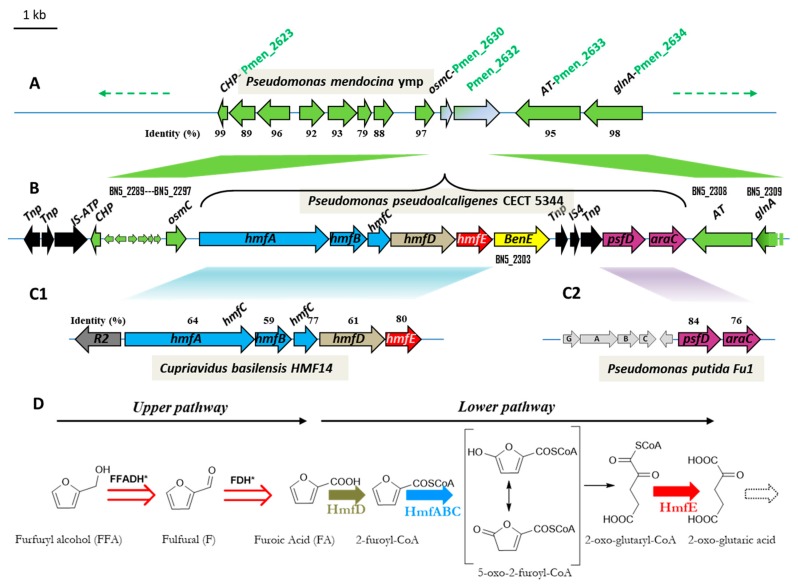
Scheme of the genetic organization of the *hmf* operon (**A**–**C**) (adapted from Ref. [9]) and predicted metabolic pathway for the assimilation of furfuryl alcohol in *Pseudomonas pseudoalcaligenes* (**D**). The *hmf* locus in *P. pseudoalcaligenes* CECT 5344 (delimited by a curly bracket, **B**) is located between BN5_2297 (*osmC*) and BN5_2308 (*AT*). The corresponding homologous genes in *Pseudomonas mendocina* ymp (*Pmen_2630-2633*) are consecutive in its genome (**A**) and the syntney of the homologous genes is conserved both upstream and downstream the *hmf* locus (green arrows). The *hmf* locus contains genes homologous to that described in the context of furfural degradation in *Cupriavidus basilensis* [24] (**C1**) and in *Pesudomonas putida* [25] (**C2**). The black arrows (panel **B**) represent genes involved in the transposition of mobile genetic elements. (**D**) Proposed pathway for the assimilation of FFA in *P. pseudoalcaligenes* R1. FFADH and FDH are furfuryl alcohol and furfural dehydrogenase, respectively. * FFADH and FDH could be the same enzyme, whose coding genes are unknown in *P. pseudoalcaligenes*. Both the oxidation of FFA and HMF in *C. basilensis* (not shown) converge in FA constituting the upper pathway. The lower pathway, encoded by the *hmfABCDE* operon (**C1**), is a series of reactions transforming FA into 2-oxo-glutaric acid. HmfA is a furoyl-CoA synthetase and HmfABC a furoyl CoA dehydrogenase. The transformation of 5-oxo-2-furoyl-CoA into 2-oxo-glutaril-CoA has no assigned gene and can be a spontaneous reaction or could be catalysed by an unspecific lactone hydrolase. Finally, HmfE was proposed to be the thioesterase rendering 2-oxo-glutaric acid from 2-oxo-glutaryl CoA [24].

**Figure 2 genes-10-00499-f002:**
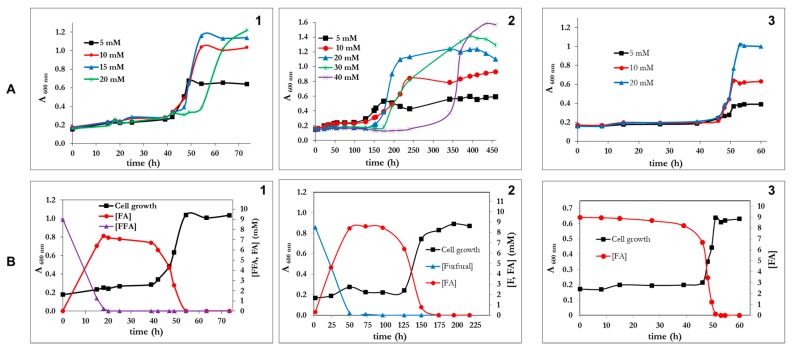
(**A**) Growth curves of *Pseudomonas pseudoalcaligenes* CECT 5344 R1 with different concentrations of furfuryl alcohol (panel A1), furfural (panel A2) or furoic acid (panel A3). (**B**) Growth curves and concentrations of furanic intermediates in the culture media of *P. pseudoalcaligenes* using as a carbon source either 5 mM furfuryl alcohol (panel B2), furfural (panel B2) or furoic acid (panel B3). Three independent experiments gave similar results.

**Figure 3 genes-10-00499-f003:**
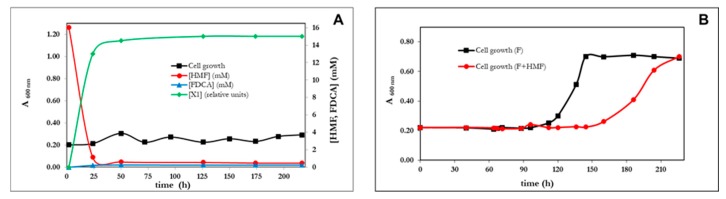
Biotransformation of HMF by *P. pseudoalcaligenes* CECT 5344 R1. (**A**) Cell growth (black line), concentration of HMF and FDCA (mM), and an unknown metabolite in cell cultures of *P. pseudoalcaligenes*. (**B**) Effect of HMF (5 mM) on the cell growth of *P. pseudoalcaligenes* at the expense of 10 mM furfural (F+HMF) in comparison with the cell growth with 10 mM furfural (F, black line). Similar results were obtained in three different experiments.

**Figure 4 genes-10-00499-f004:**
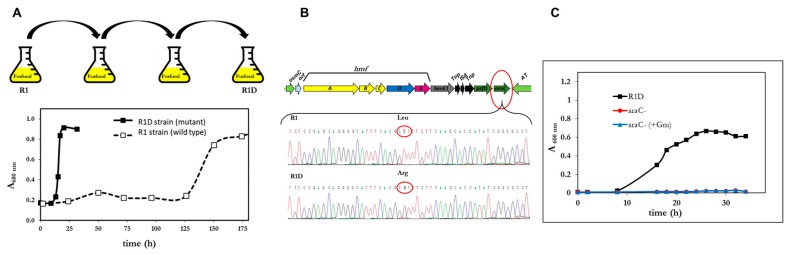
Role of *araC* in the assimilation of furfural by *P. pseudoalcaligenes*. (**A**) Scheme of the selection process (upper panel), and growth curve (lower panel) of the R1D mutant in comparison with the wt strain (R1), in media with furfural (10 mM) as the sole carbon source. (**B**) Scheme of the point mutation detected in the *araC* gene of the R1D mutant. (**C**) Growth curve of the R1D mutant and its derived mutant generated by the inactivation of the *araC** gene by insertion of the gentamicin resistance gene (*aacC1*) by double recombination. Similar results were obtained in three different experiments (panels A and C).

**Figure 5 genes-10-00499-f005:**
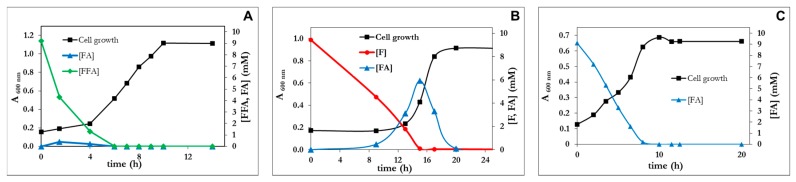
Growth curves of *P. pseudoalcaligenes* CECT 5344 R1D with furfuryl alcohol (**A**), furfural (**B**) or furoic acid (**C**) as the sole carbon sources (10 mM). The concentration of furanic intermediates in the culture media was measured by HPLC at the indicated times. Each experiment was done in triplicate giving similar results.

**Figure 6 genes-10-00499-f006:**
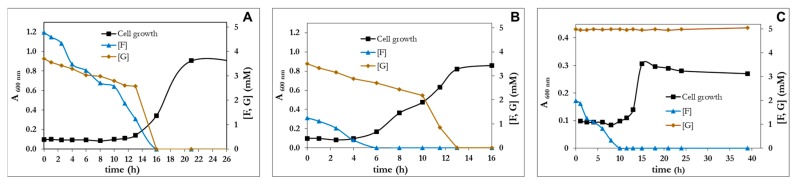
Cell growth of *P. pseudoalcaligenes* CECT 5344 R1D with glucose 5 mM, supplemented with furfural 5 mM (**A**) or 2 mM (**B**) as carbon sources. (**C**) Cell growth of *P. pseudoalcaligenes* CECT 5344 R1D *edd*^−^ with furfural (2.5 mM) and glucose (5 mM) as carbon sources. Glucose and furfural concentration were determined at the indicated times. Similar results were obtained in three different experiments.

**Table 1 genes-10-00499-t001:** Primers utilized in this work.

Name	Sequence (5′→3′)	Description
edd_9_U	CCGCGTTGTTGAAGTGACCGA	Amplification of *edd* gene
edd_730_L	GCCCGGATCCATGAAGGAAGC	Amplification of *edd* gene (BamHI).
edd_1140_U	CTACACCCGGGATCCCTTCCT	Amplification of *edd* gene (BamHI).
edd_1737_L	CGCATGAAGGCGAACAACTCG	Amplification of *edd* gene
araC_157_U	CCGGGGCCCGACCGCAATGTG	Amplification of *araC* gene (ApaI)
araC_823_L	GGGCCCGTCCACTACCCGCTG	Amplification of *araC gene* (ApaI)

**Table 2 genes-10-00499-t002:** Growth parameters of *P. pseudoalcaligenes* CECT 5344 R1 (wt) and the R1D mutant in comparison to bibliographic data.

Bacterial Strain	Growth Rate (h^−1^)			Lag Time (h)			Maximal A_600 nm_		
	FFA ^1^	F ^2^	FA ^3^	FFA	F	FA	FFA	F	FA
*P. pseudoalcaligenes* R1	0.14	0.02	0.19	40	125	48	0.68	0.59	0.39
*P. pseudoalcaligenes* R1D	0.34	0.29	0.34	2	13	<1	0.72	0.55	0.41
*C. basilensis* HMF14 ^4^	0.22	0.22	0.29	-	-	<1	1.1	1.09	0.99
*P. putida* S12 ^5^	-	0.3	-	-	-	-	-	-	-

^1^ Furfuryl alcohol, ^2^ Furfural or ^3^ furoic acid (5 mM). ^4,5^ Extracted form [16] and [24], respectively.

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
