# Peer review of "A Case of Adaptive Laboratory Evolution (ALE): Biodegradation of Furfural by Pseudomonas pseudoalcaligenes CECT 5344"

_genes, 2019, doi:10.3390/genes10070499_

Round 1
Reviewer 1 Report
Major comments:
1. English proofreading should be done for most parts of the manuscript. Some examples (by far not all):
o L 18: ‘are byproducts of sugar generation/production during the hydrolysis’
o L 19: ‘inhibits the yeasts fermentation’
o L 22: ‘did not showed’
o L 36: ‘needs international agreement’
o L 38: ‘either by avoiding their production or by mitigation their impact’
o L 41 ‘CECT 5344 bacterium was isolated’
o L 46: ‘has also made it possible’
o L 62: ‘furfural become is oxidized’
o L 83: ‘was obtained from after 4 serial transfers’
o L 86: ‘Cells cultures’
o L 246: ‘as C-source, serial dilutions of a culture of P. pseudoalcaligenes CECT 5344 R1 spread on solid minimal medium with furfural or FA gave colonies’
o and many more
2. In general, the manuscript should describe and summarize the functions of the furfural degradation genes more precisely as the focus of the work is (or seems to be) the evolved araC gene in strain R1D.
L 68: The gene benE is mentioned and the family as well, but the authors should clarify if this gene plays a role in the degradation of furfural or not.
3. Figure 1 is not well explained and should be made more comprehensible.
The genes A–E should be named hmfA–hmfE and the function of the single genes should be explained in the caption as well as in the text.
The authors should explain why Pseudomonas mendocina ymp is depicted in the figure and what conclusions can be drawn from it. The current presentation is very confusing. Much more, the authors should reconsider the necessity of presenting the genetic organization of this strain, as it does not degrade furfural.
It is not clear whether the ‘D’ in panel D is homologous to psfD (which it seems to be, then why not name it ‘psfD’?) or to ‘D’ in panel B (which might be hmfB). Furthermore, the authors should explain the functions of the genes ‘A’, ‘B’, ‘C’, and ‘G’ of panel D, since it is not apparent if they are homologues of hmfABCG or not. If they are, they should be named accordingly and the similarities should be given in percent.
4. Many growth curves are presented but the preparation of the respective cultures is not explained sufficiently. The authors should be more precise in describing these experiments, as the condition of the preculture and the preparation of the final culture have decisive influence on the duration of the lag phase as well as on the growth rate. Furthermore, a clearer statement could have been made if the experiments had been conducted in triplicate.
5. The authors should definitely present and discuss their results of the enzymatic assays for FDH and FFADH described in materials and methods including e.g. graphs showing the change of absorbance at 340 nm and 552 nm, respectively. In addition, the indication ‘appropriate amount of cell-free extract’ (L 111) is way too vague and should be clarified.
6. The mutagenesis experiments are not well explained and it is difficult to understand what has been done and why. The authors should rewrite this part and make it more comprehensible. For example, DNA maps showing which DNA regions have been amplified (including primers and names of genes etc.) and showing the constructed plasmids should be included. The compositions of the PCR reactions are missing completely.
Minor comments:
1. L 22: R1D (not RID)
2. L 40: ‘respects the environment’ this is a very unspecific statement and should be omitted
3. L 43: ‘although’ makes no sense in terms of content. Sentence should be rewritten
4. L 44: jewelry production?
5. L 88: ‘Sodium acetate’ is mentioned as substrate in materials and methods only, but no results are obtained, therefore, it should be omitted.
6. The degree of homology between the genes (or proteins) of strain R1D and the already described furfural degrading strains should be given in percent (% identity/% positives) for each individual gene/protein not only in figure 1, but also in the text.
7. L 58: ‘some microorganisms’, and L 61: ‘is limited to gram-negative aerobic bacteria, with a few notable exceptions’: The authors should name all described strains that are capable of degrading furfural and/or HMF, or at least give several distinct examples (more then C. basilensis HMF14 and P. putida Fu1) so that the reader can get an overview.
8. The authors should edit Figures 2, 3, 4, 5, and 6, since they are difficult to read because the font is too small.
9. L 233: The authors should clarify which genes/proteins are meant by ‘DH1 and DH2’ as they are not specified previously.
Author Response
Answers reviewer 1
Note: The answers, in italic and courier new after every question.
Major comments:
1. English proofreading should be done for most parts of the manuscript. Some examples (by far not all):
o L 18: ‘are byproducts of sugar generation/production during the hydrolysis’
o L 19: ‘inhibits the yeasts fermentation’
o L 22: ‘did not showed’
o L 36: ‘needs international agreement’
o L 38: ‘either by avoiding their production or by mitigation their impact’
o L 41 ‘CECT 5344 bacterium was isolated’
o L 46: ‘has also made it possible’
o L 62: ‘furfural become is oxidized’
o L 83: ‘was obtained from after 4 serial transfers’
o L 86: ‘Cells cultures’
o L 246: ‘as C-source, serial dilutions of a culture of P. pseudoalcaligenes CECT 5344 R1 spread on solid minimal medium with furfural or FA gave colonies’
o and many more
Answer: Thank you for your corrections, which have been included in the new version as well as those suggested by an English teacher.
2. In general, the manuscript should describe and summarize the functions of the furfural degradation genes more precisely as the focus of the work is (or seems to be) the evolved araC gene in strain R1D.
Answer: Effectively, the work is focused in the role of araC. Nevertheless, as required by the referee in the “major comment 3”, the function of the genes is now described in the figure caption. The role of the genes has been included in the text (Introduction section, L76-83).
L 68: The gene benE is mentioned and the family as well, but the authors should clarify if this gene plays a role in the degradation of furfural or not.
Answer: We know that this gene codes a furoate transporter. This gene is not essential for FA assimilation at high FA concentrations, but a mutant in this gene shows a clear delay respect to the wt at low FA concentrations. The reason for this probably lies in the fact that there are paralogous genes in the genome. These results are out of the scope of this manuscript. Nevertheless, as advised by the referee, we have included in the discussion section a new phrase suggesting the function of this gene (L531-554).
3. Figure 1 is not well explained and should be made more comprehensible.
The genes A–E should be named hmfA–hmfE and the function of the single genes should be explained in the caption as well as in the text.
The authors should explain why Pseudomonas mendocina ymp is depicted in the figure and what conclusions can be drawn from it. The current presentation is very confusing. Much more, the authors should reconsider the necessity of presenting the genetic organization of this strain, as it does not degrade furfural.
Answer: Figure 1 has been completely reformed and the caption written according to the referee suggestions. The reason for including P. mendocina is that it is the closer relative of P. pseudoalcaligenes (more than 3200 orthologous genes). The figure illustrates that the hmf locus resembles an island in between genes highly homologous among both strains, that are consecutive in P. mendocina genome.
It is not clear whether the ‘D’ in panel D is homologous to psfD (which it seems to be, then why not name it ‘psfD’?) or to ‘D’ in panel B (which might be hmfB). Furthermore, the authors should explain the functions of the genes ‘A’, ‘B’, ‘C’, and ‘G’ of panel D, since it is not apparent if they are homologues of hmfABCG or not. If they are, they should be named accordingly and the similarities should be given in percent.
Answer: The names of the genes and their function has been included. Panel C has been divided into C1 and C2, and panel D is now employed to show the metabolic pathway. We hope these changes clarify the message of the figure 1.
4. Many growth curves are presented but the preparation of the respective cultures is not explained sufficiently. The authors should be more precise in describing these experiments, as the condition of the preculture and the preparation of the final culture have decisive influence on the duration of the lag phase as well as on the growth rate. Furthermore, a clearer statement could have been made if the experiments had been conducted in triplicate.
Answer: The experiments were conducted in triplicate. This has been introduced in the legend of every figure. The composition and origin of the inoculum has been specified in the Materials and Methods section.
5. The authors should definitely present and discuss their results of the enzymatic assays for FDH and FFADH described in materials and methods including e.g. graphs showing the change of absorbance at 340 nm and 552 nm, respectively. In addition, the indication ‘appropriate amount of cell-free extract’ (L 111) is way too vague and should be clarified.
Answer: The protocol for measuring the enzymatic assays described in this manuscript are not original. Therefore, perhaps the paper will be to long if we include the graphs showing the individual activities, as well as the elution profiles of them form the cromatographies, and the optimum pH and temperatures of both enzymatic activities, and the induction of the activities in different culture media. We think that the essential information is indicated in the test of the results section (L355-362). Here we have made clear that FFADH and FDH are shown in Figure 1. Concerning the amount of enzyme used in the assays, this depend on the specific activity of every cell-free extract. Nevertheless, we have included a sentence indication that the appropriate amount of cell-free extract was 50-100 ml cell-free extract into 1 ml assay mixture (0.5-1 mg protein, approximately)(L167).
6. The mutagenesis experiments are not well explained and it is difficult to understand what has been done and why. The authors should rewrite this part and make it more comprehensible. For example, DNA maps showing which DNA regions have been amplified (including primers and names of genes etc.) and showing the constructed plasmids should be included. The compositions of the PCR reactions are missing completely.
Answer: The mutagenesis experiments have been rewritten and DNA maps showing DNA regions amplified and the scheme showing the construction of plasmids have been included as supplementary figures (S1 and S2 at the end of the revised version).
Minor comments:
1. L 22: R1D (not RID)
Answer: ok
2. L 40: ‘respects the environment’ this is a very unspecific statement and should be omitted
Answer: ok
3. L 43: ‘although’ makes no sense in terms of content. Sentence should be rewritten
While has been used, instead of although.
4. L 44: jewelry production?
Answer: Yes we refers to the jewellery industry. It has been corrected.
5. L 88: ‘Sodium acetate’ is mentioned as substrate in materials and methods only, but no results are obtained, therefore, it should be omitted.
Answer: Acetate was used as carbon source for studying the inducibility of FFADH and FDH enzymatic activities (L359). Now I realise that we said acetic acid instead of acetate. This has been corrected.
6. The degree of homology between the genes (or proteins) of strain R1D and the already described furfural degrading strains should be given in percent (% identity/% positives) for each individual gene/protein not only in figure 1, but also in the text.
Answer: OK, this information has been included in the text, concretely in the introduction (L85-90, L93-94).
7. L 58: ‘some microorganisms’, and L 61: ‘is limited to gram-negative aerobic bacteria, with a few notable exceptions’: The authors should name all described strains that are capable of degrading furfural and/or HMF, or at least give several distinct examples (more then C. basilensis HMF14 and P. putida Fu1) so that the reader can get an overview.
Answer: There are some extensive review concerning FA and HMF degradation. In this context, we refers to fungi. Therefore, the phase has been changed as follows: “The variety of furanic compounds degrading species is limited mostly to Gram-negative aerobic bacteria and some Gram positives [17], with a few exceptions including fungi [14] (L72-73).
8. The authors should edit Figures 2, 3, 4, 5, and 6, since they are difficult to read because the font is too small.
Answer: All the fonts in these figures have been increased 2 points.
9. L 233: The authors should clarify which genes/proteins are meant by ‘DH1 and DH2’ as they are not specified previously.
Answer: DH1 and DH2 correspond to FFADH and FDH, respectively. We named them in this way in the previous version because their corresponding coding genes are unknown. This has been modified now in order to homogenise the nomenclature.

Reviewer 2 Report
The manuscript presented by Igeño et al shows an interesting example of how a new pathway can be assembled by HGT followed of genetic adaptation. Igeño et al describe the furfural degradation pathway in P. pseudoalcaligenes and how a genetic adaptation is requires in order to optimizing furfural metabolism. Additionally, by blocking the glucose metabolism the authors engineer an interesting bacterial strain allowing furfural removal while avoiding glucose metabolism. The work is well done and the conclusion are overall well supported.
Minor comments:
1) My main concern is about the potential HGT origin of the furfural cluster in P. pseudoalcalintes. Beyond of the syntheny observed with the genes cluster of C. basiliensis and P. putida Fu1, can the authors provide additional evidences supporting this hypothesis? e.g., genetic features such as CG, codon usage and so on.
2) The authors identify just only single mutation between strains R1 and R1D, however they found several different evolved strains harbouring the capability to use FA as carbon source. In how many of these strains the gene encoding for the AraC-type regulator was sequenced?. It would be interesting to know whether only this single mutation provide the phenotype observed in the strain R1D.
Author Response
Answers reviewer 2
Note: The answers, in italic and courier new after every question.
Comments and Suggestions
The manuscript presented by Igeño et al shows an interesting example of how a new pathway can be assembled by HGT followed of genetic adaptation. Igeño et al describe the furfural degradation pathway in P. pseudoalcaligenes and how a genetic adaptation is requires in order to optimizing furfural metabolism. Additionally, by blocking the glucose metabolism the authors engineer an interesting bacterial strain allowing furfural removal while avoiding glucose metabolism. The work is well done and the conclusion are overall well supported.
Minor comments:
1) My main concern is about the potential HGT origin of the furfural cluster in P. pseudoalcalintes. Beyond of the syntheny observed with the genes cluster of C. basiliensis and P. putida Fu1, can the authors provide additional evidences supporting this hypothesis? e.g., genetic features such as CG, codon usage and so on.
Answer: We agree that we can not conclude the origin of the cluster. Accordingly, instead of saying that the genes were transferred form Cupriavidus and Pseudomonas, we say that the transferred genes are “genes homologous to the catalytic genes for the assimilation of FA described in C. basilensis…” (L103-104) and L567.
The %GC of the individual genes in the locus has been analysed and included in the discussion section (L558-564). In this island the %GC varies respect the average, thus suggesting the HGT event from two different origins took place.
2) The authors identify just only single mutation between strains R1 and R1D, however they found several different evolved strains harbouring the capability to use FA as carbon source. In how many of these strains the gene encoding for the AraC-type regulator was sequenced?. It would be interesting to know whether only this single mutation provide the phenotype observed in the strain R1D.
Answer: Please take into account that we compared the complete genome sequence of the wt and the tanscriptome of R1D. R1D was not obtained from a single colony, but by selecting the fast growing strain after several reinoculations (4 times). Then, we re-sequenced araC in R1D from a single colony and the same mutation was there. Since we did not sequenced the entire genome we cannot conclude absolutely that this single mutation is the responsible for the phenotype. Nevertheless, the fact that mutating the mutated araC in R1D recover back the phenotype of the wt strain strongly suggest that only this single mutation provide the phenotype observed in the strain R1D.

Round 2
Reviewer 1 Report
The revised version includes many important corrections. Still, some things need to be improved.
Minor comments:
1. L 18: It should read either ‘generation’ or ‘production’.
2. L. 44: ‘jewellery’ is BE, ‘jewelry’ is AE
3. L. 207 ff: Explaining the PCR reaction conditions, the authors should specify the final reaction volume for 0.5 U Taq DNA polymerase and 2 µL buffer (probably 20 µL?) or refer to one volume (e.g. 1 µL buffer per 10 µL reaction volume).
4. Figure 1, panel C2: ‘D’ should be renamed ‘psfD’
5. In the version provided, Figures 2, 3, 4, 5, and 6 are still difficult to read due to the small font. The authors should take this into account in the final figure size.
6. L. 249 ff: ‘It is remarkable the fact that with at the same concentration of furanic compound, the maximal maximum growth was highest with FFA, followed by F and finally FA.’
7. L. 106 ff: ‘Materials and methods’ state that the inoculum for growth curves were overnight cultures diluted 1:10. Figure 2 shows that growth of the cultures start after more then 40 hours. On this basis, the use of an overnight culture makes no sense at all since this culture had not even started to grow. Again, the authors should explain more precicely the real conditions of the precultures used including the substrate applied, and the preparation of the respective cultures used in the experiments.
8. L. 346-347: One of these sentences is sufficient.
9. L. 348: ‘capable of to assimilate’
10. L. 371: ‘maximum’ instead of ‘maximal’
11. L. 388: ‘pseudoalcaligenes’
12. L. 435: ‘in of P. putida KT(2440?) it was detected a single point mutation was detected causing’
13. L. 439: ‘and another one homologous to’
14. L. 446: ‘in is flanked’
15. L. 465: P. putida KT or P. putida KT2440?
16. L. 470: A subject is missing in sentence, ‘In this point converges the assimilation of HMF and furfural in C. basilensis HMF14‘ and should be included
Author Response
Please see the attachmen.
